# Research on the Behavior Recognition of Beef Cattle Based on the Improved Lightweight CBR-YOLO Model Based on YOLOv8 in Multi-Scene Weather

**DOI:** 10.3390/ani14192800

**Published:** 2024-09-27

**Authors:** Ye Mu, Jinghuan Hu, Heyang Wang, Shijun Li, Hang Zhu, Lan Luo, Jinfan Wei, Lingyun Ni, Hongli Chao, Tianli Hu, Yu Sun, He Gong, Ying Guo

**Affiliations:** 1College of Information Technology, Jilin Agricultural University, Changchun 130118, China; muye@jlau.edu.cn (Y.M.); 20231337@mails.jlau.edu.cn (J.H.); 20231236@mails.jlau.edu.cn (H.W.); 20231226@mails.jlau.edu.cn (H.Z.); 20231225@mails.jlau.edu.cn (L.L.); weijinfan@mails.jlau.edu.cn (J.W.); nilingyun@mails.jlau.edu.cn (L.N.); chaohongli@mails.jlau.edu.cn (H.C.); hutianli@jlau.edu.cn (T.H.); sunyu@jlau.edu.cn (Y.S.); 2Jilin Province Agricultural Internet of Things Technology Collaborative Innovation Center, Changchun 130118, China; 3Jilin Province Intelligent Environmental Engineering Research Center, Changchun 130118, China; 4Jilin Province Colleges and Universities and the 13th Five-Year Engineering Research Center, Changchun 130118, China; 5College of Information Technology, Wuzhou University, Wuzhou 543003, China; lishijun@jlau.edu.cn; 6Guangxi Key Laboratory of Machine Vision and Intelligent Control, Wuzhou 543003, China

**Keywords:** cattle behavior, YOLOv8, behavior recognition, MCFP, LMFD, lightweight model

## Abstract

**Simple Summary:**

Cattle behavior recognition is an important field in animal husbandry. It can be used to understand the health status, emotions and needs of cattle. In this paper, an accurate and lightweight behavioral multi-detection model is proposed, which is adapted to real weather conditions. An innovation in the head, neck, detection head and loss function of the model is proposed, which improves the accuracy of behavior detection in cattle, and greatly reduces the number of parameters and calculations. It not only has high accuracy in recognition tasks, but is also very friendly to edge devices. This gives breeders insight into cattle behavior, helping them to better manage their herds, improve breeding efficiency and ensure the health and welfare of their cattle.

**Abstract:**

In modern animal husbandry, intelligent digital farming has become the key to improve production efficiency. This paper introduces a model based on improved YOLOv8, Cattle Behavior Recognition-YOLO (CBR-YOLO), which aims to accurately identify the behavior of cattle. We not only generate a variety of weather conditions, but also introduce multi-target detection technology to achieve comprehensive monitoring of cattle and their status. We introduce Inner-MPDIoU Loss and we have innovatively designed the Multi-Convolutional Focused Pyramid module to explore and learn in depth the detailed features of cattle in different states. Meanwhile, the Lightweight Multi-Scale Feature Fusion Detection Head module is proposed to take advantage of deep convolution, achieving a lightweight network architecture and effectively reducing redundant information. Experimental results prove that our method achieves an average accuracy of 90.2% with a reduction of 3.9 G floating-point numbers, an increase of 7.4%, significantly better than 12 kinds of SOTA object detection models. By deploying our approach on monitoring computers on farms, we expect to advance the development of automated cattle monitoring systems to improve animal welfare and farm management.

## 1. Introduction

Cattle behavior recognition technology provides breeders with insight into the health, mood and needs of their cattle, which helps them manage their herd more effectively, improve breeding efficiency and ensure the health and welfare of their cattle. Such technological progress is of great significance for ensuring human food safety and promoting sustainable development in agricultural science and technology.

Currently, there are two types of livestock behavior detection methods: contact [1,2,3] and non-contact methods [4,5,6]. The contact methods refer to the installation of wearable sensors on livestock; in the case of cattle, wearable devices can cause stress to the animals. Although smart sensors are usually designed to be non-invasive, methods such as collars or ear tags can still cause a degree of damage to the animal’s fur, triggering the animal’s stress response and interfering with other subsequent forms of detection. In addition, wearable devices are expensive, easy to damage and difficult to reach, and the physical movement of animals will also cause the sensor to reposition itself, making subsequent research observations biased.

In recent years, with the rapid development of the State of the Art (SOTA) generative model, single-stage object detection algorithms such as You Only Look Once (YOLO) are constantly improving. Making large-scale contact-free detection of animals based on computer vision possible, the YOLO algorithm has been proven to be superior to other algorithms in the field of target detection and recognition, and has been widely recognized in the industry. Zhu et al. improved the convolutional attention and feature pyramid of YOLO v5 to improve the recognition accuracy of the model [7]. However, the application scenario of this technology is relatively specific, mainly limited to sunny weather conditions with sufficient sunlight. However, in real-world applications, weather conditions are often fickle, so the scope of this approach is relatively limited. Qiao et al. proposed the YOLOv5-ASFF target detection model, designed to detect the individual, head and leg of cattle with an accuracy of 96.2% [8]. However, the detection scenario is single, individual body parts are detected separately, and the limiting factors are too large to conduct joint characterization of each sign. Inevitably, the early warning of the health status of cattle is not timely. In previous studies, researchers mainly focused on the accuracy rate while ignoring the network’s training speed and number of parameters. However, they usually increased the training speed at the cost of increasing the number of parameters. The huge number of parameters and amount of calculations in the object detection algorithms make it difficult to deploy on resource-limited devices. In order to solve the above problems and to better detect the daily behavior state of cattle, this paper improved the model by using YOLOv8 [9], which can recognize both still images and video data [10]. In this paper, we proposed a new, lighter, more efficient and more accurate bovine posture recognition model.

The contributions of this article are as follows:Inner-MPD IoU Loss is proposed in this paper, which can handle the fine details of cattle, solve the problems of bounding box regression and data set imbalance, improve the computational efficiency and enhance the model interpretation by using the chain rule.The novel structure of the Multi-Convolutional Focused Pyramid (MCFP) module is innovatively proposed. Through the pyramid-type diffusion mechanism, the module enables various scale features to be integrated into rich contextual information, so that the network can explore and learn in depth the detailed features of the cow in different states.Design a new Detection Head. The Lightweight Multi-Scale Feature Fusion Detection Head (LMFD) is designed to take full advantage of deep separable convolution without increasing computational complexity. This means that our model can achieve richer expressiveness and stronger feature representation while maintaining computational efficiency.

## 2. Materials and Methods

### 2.1. Materials

#### 2.1.1. Data Source

The cattle dataset in this study consists of two parts: one part is from the NWAFU-CattleDataset [11], which was captured by the Animal Husbandry Teaching Test Base of Northwest Agriculture and Forestry University of Yang Ling in China under field conditions using a smartphone. The other part was obtained by us in the Changchun BoYu Agricultural Cattle Training Base, and the acquisition equipment was a Canon camera (model: Canon EOS 5 D Mark II). The images were captured at 1920 × 1820 resolution and saved in JPG format, as shown in Figure 1. To further improve the complexity of the dataset, we carefully selected ten beef cattle video clips with complex backgrounds from the Pixabay website (https://pixabay.com, accessed on 23 June 2024) to enrich the diversity of the dataset.

#### 2.1.2. Data Set Construction

We selected over 2000 pictures of multiple cows in natural breeding environments and cattle with body area occlusion, carried out frame extraction processing on the video data, and set the frame rate at 15 fps. In this study, the postures of four types of cattle were marked: standing, walking, eating and lying down. The annotation process uses the LabelImg tool, and all images are annotated manually and saved in the YOLO dataset format. In order to prevent the model from overfitting, we performed image enhancement on the data, including random cropping, HSV color jitter, Gaussian noise, horizontal flip, and scaling. After enhanced processing, the data volume was increased to 5051. We then randomly divided the data set into three groups at a ratio of 8:1:1, i.e. the training, validation, and test sets. We ensured that cattle in the same environment are present in all data sets. The distribution among these data sets is shown in Table 1, and the collected data samples are shown in Figure 1.

Since it is difficult to capture weather changes in the real environment, in order to simulate the real farming environment, we selected 50% of the representative images from the training set. We created images of different weather conditions using RGB channel synthesis technology, where the intensity and location of the weather were random. Using the linear mixing method, the original image is weighted by randomly generating brightness parameters A and transmittance t to achieve a weather generation effect. The calculation formula is as follows:(1)I(x)=J(x)×t(x)+A(1−t(x))
where  x is the pixel coordinates, I(x) represents the synthesized image, J(x) represents the original image,  t(x) is the transmittance map, and A is the atmospheric light value. Figure 2 shows a composite weather image of moderate intensity.

### 2.2. Method

#### 2.2.1. Cattle Behavior Recognition-YOLO (CBR-YOLO)

YOLOv8 effectively solves the problem of information loss and resolution mismatch, but at the same time greatly increases the number of parameters, resulting in slower model training and reasoning. To solve the above problems, we propose a CBR-YOLO model, which replaces the traditional convolution of the trunk with ultra-lightweight StarNet and integrates a self-calibration module in the Spatial Pyramid Pooling in Feature Maps (SPPF) layer. It is proposed that the MCFP module fully integrates context information and designs a lightweight detection head to improve model performance. The overall model improvement method is shown in Figure 3, and the four red dotted boxes represent the improvements made in this study:

#### 2.2.2. StarNet

The traditional convolutional network of the YOLOv8 model has limitations in the high-dimensional nonlinear transformation of feature representation, while StarNet, proposed by Ma X [12], can recursively increase the implicit feature dimension and use a lightweight network to realize the spatial mapping of high-dimensional and nonlinear features.

The architecture of the StarNet network in this paper is shown in Figure 4. (a) represents the hierarchical network structure of StarNet at each stage. In this study, it is designed as a four-stage hierarchical structure. Each stage consists of a layer of 3 × 3 convolution and Star Blocks, through which down-sampling is performed, and an optimized demonstration module is used for feature extraction. In order to ensure the efficiency of the algorithm, layer normalization is replaced by batch normalization, the batch normalization layer is placed after the deep separable convolution layer, and a DW-Conv is placed at the end of each Block. This structure can be fused in the inference stage. This means that the quality of feature extraction can be significantly improved through appropriate depth and width design, thus improving the accuracy of target detection. The channel expansion factor is initialized to 4 and the width is doubled in each stage of the network. All ordinary convolution has a convolution kernel size of 3 and a step size of 2, and depth-separable convolution has a convolution kernel size of 7 and a step size of 1. When this structure is applied to a neural network and stacked through multiple layers, each layer brings an exponential increase in the complexity of the implicit dimensions. This means that we can use unsupervised learning techniques to reconstruct high-dimensional data from low-dimensional sparse representations without complex design or carefully selected hyperparameters, thereby achieving high performance while reducing the number of parameters and significantly improving inference speed, demonstrating its operational efficiency.

#### 2.2.3. SPPF-LSKA

In YOLOv8, the SPPF layer effectively improved the multi-scale object detection ability, but it had high computational complexity and could not capture all the scale details in cattle motion capture research. Therefore, the LSKA large kernel attention module proposed by Lau [13] was used in this study, which decomposed the two-dimensional convolution kernel of the depth-separable convolution into a cascaded one-dimensional kernel, allowing the direct application of the large kernel depth convolution layer in the attention module, reducing memory and computational complexity. For the convenience of comparison, we show the original SPPF structure diagram, the LSKA module structure diagram and the modified SPPF module structure diagram, as shown in Figure 5.

Specifically, d represents the expansion rate, $\frac{k}{d}$ represents the size of the convolution kernel, C is the number of input channels, and H and W represent the height and width of the feature map, respectively. Combining the formula proposed by Guo [14] for visual attention networks, the output of LSKA can be expressed as follows: Where ${\bar{Z}}^{C}$ represents the output of the deep convolution, which captures local spatial information and compensates for the grid effect of the following deep extended convolution, $Z^{C}$ representing the output of the deep convolution obtained by convolving the kernel W of size k×k with the input feature map, and the ${\bar{F}}^{C}$ is the resulting Hadamard product of the input feature map $F^{C}$ of the attention map $A^{C}$. SPPF-LSKA is connected to LSKA through a 1 × 1 convolution, and two maximum pooling layers are connected in series, and then input to a 1 × 1 convolution through a fully connected layer. In this way, the improved SPPF layer can identify and enhance the key areas in the image more effectively, making the model pay more attention to the key areas and edges of the image. Moreover, for the detailed features of high occlusion and high blur in this study, it can help us better deal with occlusion and illumination transformation, and improve robustness in complex environments.
(2)$${\bar{z}}^{c}=\underset{H,W}{\sum }W_{\left(2d-1\right)\times 1}^{C}\times \left(\underset{H,W}{\sum }W_{1\times \left(2d-1\right)}^{C}\times F^{C}\right)$$
(3)$${\bar{z}}^{c}=\underset{H,W}{\sum }W_{\left[\frac{k}{d}\right]\times 1}^{C}\left(\underset{H,W}{\sum }W_{1\times [\frac{k}{d}]}^{C}\times {\bar{Z}}^{C}\;\right)$$
(4)$$A^{C}=W_{1\times 1}\times Z^{C}\;$$
(5)$${\bar{F}}^{C}=A^{C}⊗F^{C}$$

#### 2.2.4. Multi-Convolutional Focused Pyramid Module

YOLOv8’s Feature Pyramid Network (FPN) fuses multi-scale features through residual blocks and horizontal connections. The selection of layers affects performance, and details are easily lost in dense targets and complex backgrounds. Since beef cattle motion recognition needs to capture some small targets and ensure that no details will be lost in the process of feature extraction and fusion, the MCFD module is proposed in this study. The architecture is shown in Figure 6. We adopt the idea of Adaptive Down-sampling Convolution (ADown) proposed by C.-Y. Wang [15] in YOLOv9, which can adjust the down-sampling rate adaptively according to the size of the input image. Convolution retains more information than a normal step size of 2, which allows it to use a higher down-sampling rate on smaller images, thus reducing the amount of computation. P3, P4 and P5 represent feature layers of different sizes in the backbone, and 1 × 1 convolution adjustment channels are connected to ensure that the number of channels in the feature layer is consistent.

To address the problem of detecting target scale changes, we introduce a set of parallel deep convolutions containing an inception Style module. Unlike the original functional pyramid network, it does not rely on large kernels or dilated convolutions to expand the receptive field. Instead, the initial-style deep convolution is used to extract multi-scale texture features under different receptive fields, and the scalability of PW_Conv is used to capture multi-scale context information, effectively solving the challenge of object scale change. Finally, the aggregated multi-scale features were diffused through the residual connection. The diffusion mechanism effectively propagated the features rich in context information to each detection scale, extracting global context information while minimizing the number of parameters and calculation.

Figure 7 shows the process of obtaining different receptive fields by convolution nuclei of different sizes, constructing a parallel multi-branch structure, and finally combining the feature maps of these different receptive fields.

The MCFP module effectively fuses multi-scale information using up-sampling and lateral connection techniques, as shown in Figure 8. This module passes semantic information from the higher layers to the lower layers through single-scale up-sampling and enhances the capture of local features using a 3 × 3 convolution. Through the superposition of upper and lower features, the feature map of each layer can fuse the feature information of different resolutions, thus enhancing the detection ability of the model for objects of different sizes. Specifically, high-level feature information is fused with low-level feature information through haploid up-sampling, while low-level feature information is further refined through 3 × 3 convolution to improve its sensitivity to local features. This fusion mechanism not only ensures that the model can detect small targets, but also ensures the integrity of semantic information.

#### 2.2.5. Lightweight Multi-Scale Feature Fusion Detection Head

YOLOv8 adopts a decoupling head (as shown in Figure 9), and the three detection heads adopt a double-branch structure to extract information through two 3 × 3 convolutions and one 1 × 1 convolution, respectively, which are divided into two branches of Cls classification and Box regression. After three convolutional layers, the channels are cycled through in a loop, which significantly increases the number of parameters and the computational cost of the detection head, accounting for nearly 1/5 of the overall computational cost. The vast number of parameters in the feature extraction process is bound to result in redundancy. In addition, the detection head of YOLOv8 uses a point-to-point single-scale prediction structure, which cannot effectively extract multi-scale features when dealing with cow postures.

Therefore, to solve the above problems, a lightweight multi-scale feature fusion detection head (LMFD) is proposed in this paper. The structure is shown in Figure 10. P3, P4, and P5 are small and medium scale, respectively. Feature maps of different sizes detect objects of different sizes, respectively. With the increase of network depth, feature maps become smaller, more abstract, and contain more semantic information. The feature maps of each scale level were independently calculated by a 1 × 1 Conv_GN, then shared parameters by a 3 × 3 Conv_GN, and the convolutional regression layer (Conv_Reg) and convolutional classification layer (Conv_Cls) were output. The Sigmoid Linear Unit (SiLU) activation function is used after each GN layer to maintain numerical stability. Finally, in order to deal with the problem of different detection scales of each detection head, the Scale layer is introduced to adjust the feature distribution, which alleviates the problem of reduced information interaction between channels. This not only ensures that the multi-dimensional information is fully integrated and improves the network performance, but also greatly reduces the amount of computation and the number of parameters. The YOLO series network needs to find a balance between real-time and detection accuracy, and the efficient nature of LMFD fits this need.

#### 2.2.6. Inner-MPDIoU Loss

The YOLO network series primarily calculates the loss based on the IoU loss. The Complete Intersection over Union (CIoU) used in the YOLOv8 model can provide a more comprehensive evaluation of the accuracy of the bounding box. However, it has high computational complexity and a slow convergence rate in handling occlusions and dense distributions, which is not conducive to capturing the delicate features of cattle.

In this study, the Inner-IoU Loss and MPD IoU [16,17] Loss are combined to form a new loss function, Inner-MPDIoU Loss. In cattle action recognition, the similarity measurement standard for the minimum point distance of the bounding box needs to be considered, and the MPD IoU Loss can directly minimize the distance between the predicted bounding box and the actual annotated bounding box at the top-left and bottom-right corners, solving the problem of bounding box regression and dataset imbalance. The combination of the two factors comprehensively considers the influence of multiple geometric factors to improve the stability of performance. The calculation factor of the existing bounding box regression index is shown in Figure 11. The existing bounding box regression metrics are calculated based on the Inner-MPDIoU Loss, which includes all relevant factors considered by existing loss functions, such as overlapping or non-overlapping regions, center point distance, and width and height deviation, as shown in Figure 12. The calculation process of Inner-MPDIoU Loss is as follows: the distances $d_{1}$ and $d_{2}$ represent the Euclidean distances between the coordinates of the predicted and actual bounding boxes. The term $w^{2}+h^{2}$ corresponds to the squared length of the existing diagonal of the bounding box. The ratio $\frac{d^{2}}{w^{2}+h^{2}}$ indicates the length of the redundant predicted bounding box. The Inner-MPDIoU Loss is then defined as the Intersection over Union (IoU) subtracted by the portion outside the ground truth bounding box.
(6)$$d_{1}^{2}={\left(x_{1}^{prd}-x_{1}^{gt}\right)}^{2}+{\left(y_{1}^{prd}-y_{1}^{gt}\right)}^{2}$$
(7)$$d_{2}^{2}={\left(x_{2}^{prd}-x_{2}^{gt}\right)}^{2}+{\left(y_{2}^{prd}-y_{2}^{gt}\right)}^{2}$$
(8)$$Inner-MPDIoU=IoU-\frac{d_{1}^{2}}{w^{2}+h^{2}}-\frac{d_{2}^{2}}{w^{2}+h^{2}}$$

## 3. Results and Analysis

### 3.1. Experimental Platform and Parameter Setting

In this study, the image input size was set to 640 × 640 pixels. In order to accelerate the convergence speed, the initial learning rate was set to 0.01, the stochastic gradient descent algorithm (SGD) was used for training, the weight attenuation coefficient was set to 0.0005, the momentum factor was set to 0.937, the training batch size was set to 32 times, and the number of workers was set to 12. All experiments were implemented on a Linux server, and the specific experimental environment configuration is shown in Table 2.

### 3.2. Analysis and Accuracy Evaluation of Cattle Identification Results

#### 3.2.1. Evaluation Indicators

To measure the effectiveness of our CBR-YOLO model in detecting cattle, we used a series of performance metric standards in the field of target detection.
(9)$$Precision=\frac{tp}{tp+fp}\;$$

tp, tf,fn, and fp represent the number of true positive, false positive, and false negative samples.
(10)$$mAP=\frac{1}{C}\underset{c\in C}{\sum }AP\left(c\right)\;$$

C represents a collection of object classes, ∣c∣ is the total number of categories, and AP(c) refers to the average accuracy of class c.
(11)$$FLOPs=C_{in}\times C_{out}\times K_{h}\times K_{w}\times H_{out}\times W_{out}+C_{out}\times H_{out}\times W_{out}$$

H_out and W_out are the height and width of the output of the convolutional layer, C represents the number of channels, K_h and K_w are the height and width of the convolutional kernel, respectively, and the total number of weight parameters is C_in×K_h×K_w multiplied by C_out.
(12)$$Recall=\frac{tp}{tp+fn}$$

#### 3.2.2. Comparative Experiments of Different Models

To verify the improved YOLOv8 model in detection performance, we used the same data set in this study to evaluate the performance under thirteen different models. As shown in Table 3, compared with the original YOLOv8n, the detection accuracy, mAP value, and recall rate of CBR-YOLO increased by 7.2%, 7.4%, and 8.4%, respectively, and the number of parameters and floating-point operations were reduced by 1.6 × 10^6^ and 3.9 G, respectively. The comprehensive evaluation index of YOLOv8s and YOLOv8m is higher than that of YOLOv8, but their huge number of floating point operations and parameters makes it difficult to deploy to resource-limited devices.

Regarding parameters and FLOPs, the number of parameters of the CBR-YOLO model is the smallest among all the models, and FLOPs are second only to YOLOv5n. However, the detection accuracy of YOLOv5n is far behind the CBR-YOLO model. As shown in Figure 13, other YOLO models find it challenging to balance accuracy and computation in this experiment, and redundant network architectures may lead to significant computational losses. We selected six algorithms with similar performance to comprehensively compare their detection performance. As shown in Figure 14, the farther each axis of each curve is from the intersection point, the better the metric, and the larger the area surrounded by the curve, the better the comprehensive performance of the algorithm. It can be seen that the overall indicators of the CBR-YOLO proposed in this paper are higher than those of the comparison models. The performance has been improved, and a lightweight model has also been achieved, making it more advantageous in practical applications.

Figure 15 and Figure 16 show the comparison of loss performance of CIoU Loss, Inner-IoU Loss, MPDIoU Loss, and Inner-MPD IoU Loss in this study and the test results on this experimental data set, respectively. In this study, CIoU Loss misclassified feeding behavior as walking. Both Inner-IoU Loss and MPDIoU Loss failed to detect walking behavior. Although other behaviors were correctly identified, the precision of the bounding boxes was notably low. For instance, in the first column of the images, the accuracy of detecting the standing posture of the cow on the left was 68%, 79%, 81%, and 87%, respectively. Similarly, in the second column, the accuracy of detecting the feeding behavior of the cow on the right was 67%, 55%, 64%, and 69%, respectively. In summary, the Inner-MPD IoU Loss adopted in this study has the best performance, which can better capture the matching degree of splicing edges, and has the best positioning accuracy for the recognition of different cattle poses.

Figure 17 shows the detection performance of the four models with the highest comprehensive indexes on different behaviors of cattle under different scenarios. The solid line zoom box represents the correct detection of beef cattle behavior, and the dashed line zoom box represents the error detection and missed detection. It can be seen that both Faster R-CNN and YOLOv8n have incorrect detection or missed detection in different weather. YOLOv8s and CBR-YOLO accurately detected various behaviors, but the detection accuracy of YOLOv8s is not as good as that of CBR-YOLO. Table 4 and Table 5 show these four models’ precision and mAP indicators for different cattle behaviors. It can be seen from the table that compared with other models, the CBR-YOLO model has a better recognition effect on the detection of cattle behaviors in different weather changes in complex scenarios.

### 3.3. Ablation Experiment

#### 3.3.1. The Influence of the Improved Module on the Algorithm

The proposed CBR-YOLO model is based on YOLOv8n and is optimized by replacing the loss function and introducing StarNet, LSKA, MCFP, and LMFD. In order to evaluate the performance of each optimization module, an ablation experiment was conducted using the variable control method. Training and testing were carried out on the same data set and training parameters, and the results are shown in Table 6.

It can be seen that after the introduction of Inner-MPD IoU, the detection effect was significantly improved, and the peaks of AP_lying_, AP_standing_, AP_eating_, and AP_walking_ increased by 1.2%, 0.9%, 1.7%, and 1.4%, respectively. After successfully optimizing the detection performance of small targets and considering the uniqueness of cattle behaviors, we incorporated the LSKA module into the SPPF layer of the backbone network to achieve self-calibration and fusion of features. Table 7 shows the performance comparison data of the LSKA module at different positions in the model. This strategy addresses the challenge of channel features of different scales in a complex background. We selected the MCFP module compared to YOLOv8n + Inner-MPD IoU, and adding MCFP increased the overall accuracy by 4.3%. Finally, the LMFD lightweight detection head was introduced to suppress redundant information through the cross-dimensional effect. The floating-point number of the model changed from 6.1 G to 5.2 G, and the number of parameters decreased by more than 300,000, while the mAP value increased from 90.1% to 90.2%.

To sum up, the various optimizations of YOLOv8n in this study improved the accuracy of detecting cattle behaviors. Table 8 shows the experimental comparison between the original YOLOv8 detection head and the LMFD detection head. The “√” symbol stands for use The results show that the number of parameters of the LMFD detection head is 85% less than that of the original YOLOv8, the floating-point number decreased from 8.9 G to 6.7 G, and the accuracy value changed from 83.3% to 84.9%, ensuring the average accuracy of the model while significantly reducing the computational complexity, which is of practical significance for deployment on resource-constrained development boards.

#### 3.3.2. Heat Map Visualization Analysis

In order to visually demonstrate the comparison before and after model optimization, Gradient-Weighted Class Activation Mapping [25] was used to visualize the output layer of small objects of YOLOv8n and CBR-YOLO. As shown in Figure 18, when YOLOv8n was not improved, the model was affected by the double influence of other complex backgrounds and other cattle. It could not pay good attention to the cattle’s behavior and captured less of their behaviors. However, CBR-YOLO can make the network pay more attention to the detailed areas, indicating that the optimization operation can enable the network to make full use of the context information to capture the movements and states of cattle, extract the key features of behaviors more accurately, and at the same time suppress other irrelevant background interference, thereby improving the recognition accuracy of cattle behaviors.

These results demonstrate the effectiveness of our proposed method for improving the accuracy and precision of the YOLOv8n cattle behavior detection model. The Grad-CAM visualization provides insight into the regions of the image that the model is used to predict and shows that our proposed approach is better able to focus on key features of cattle.

#### 3.3.3. Visualization of Feature Map

In order to gain a deeper insight into the function of each module in the model, this study uses visual feature maps to delineate the influence of different modules in detail. Figure 19 shows the feature effects of the output of the C2f layer of the YOLOv8n model and the StarNet layer of the CBR-YOLO. Although the original model can extract the bovine contour, the feature map resolution is low and the details are fuzzy. Especially when capturing the action, the edge fusion effect is not good, and the recognition results are not clear enough. The CBR-YOLO model can effectively integrate contextual information, carefully integrate surface details, filter background interference, and highlight the bovine subject, making subsequent feature observation and analysis more convenient. Overall, these results demonstrate the effectiveness of each of the proposed improvements to the cattle detection performance of our YOLOv8 model. In particular, Inner-MPD IoU and MCFP are effective techniques for improving the performance of object detection models, while LMFD is an important technique for improving the robustness and deployment flexibility of deep learning models.

## 4. Conclusions

This study proposes an innovative modern livestock cattle behavior monitoring technology that breaks through the dependence on wearable sensors, and achieves effective detection across scenarios and multiple targets by directly identifying images and videos of individual cattle. This method improves the operational efficiency of cattle farms and significantly reduces resource waste.

Based on the advanced YOLOv8 framework, we developed the CBR-YOLO model, which uses the random weather synthesis algorithm to approach the most realistic breeding environment, and adopts the Inner-MPD IoU to replace the traditional CIoU, effectively solving the sensitivity problem of small target positioning. In addition, we replaced the YOLOv8 backbone network with StarNet and designed a novel MCFP module, which can efficiently extract multi-scale context information and promote the deep fusion of high-level features with low-level features, thus enhancing the expression ability of low-level features and significantly improving the detection performance of cattle behavior at different scales.

Furthermore, we introduced the LMFD module, which significantly reduces the redundancy of the model, enabling the CBR-YOLO model to achieve higher accuracy and efficiency while remaining lightweight. In terms of accuracy metrics, the model demonstrates excellent performance, effectively detecting all four behavior patterns of cattle and showing comprehensive advantages in comprehensive performance evaluation.

The YOLO series of models are an important representative in the field of target detection, achieving a good balance between real-time performance and accuracy. However, CBR-YOLO has undergone more comprehensive optimization in terms of network architecture, hyperparameters, and training strategies, making it better suited for behavior recognition in airport scenarios.

This paper mainly focuses on the recognition and classification of cattle behavior, laying a foundation for precision livestock management. However, individual identification of cattle is equally important for fine-grained individualized management. In future research, we will combine deep learning, computer vision, and other technologies to explore the extraction and recognition of biological features such as cattle facial features and body shape, achieving precise identification of cattle individuals. By combining behavior recognition and individual identification, we can gain a deeper understanding of cattle health and productivity, providing a more comprehensive solution for intelligent livestock farming.

Additionally, we will broaden our research scope to encompass more complex scenarios, including tracking and analyzing the behavior of cattle, in order to gain a deeper understanding of their patterns and characteristics. We intend to explore more of the behavior of livestock using transfer learning, this series of work aims to facilitate the advancement in animal detection and comprehension. The model is sufficiently efficient and lightweight to be significant for future deployment in real-time dynamic scenarios, on devices with limited resources.

## Figures and Tables

**Figure 1 animals-14-02800-f001:**
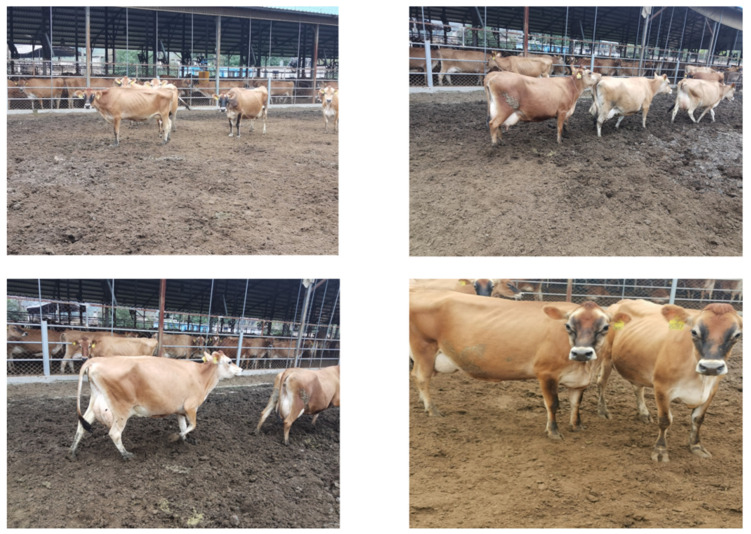
Samples of the data augmentation.

**Figure 2 animals-14-02800-f002:**
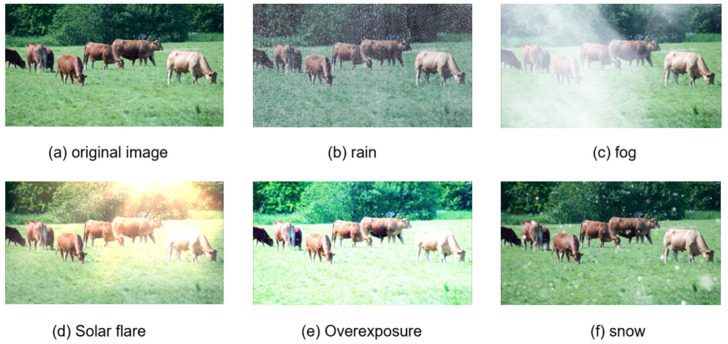
A random image of beef cattle was selected for weather synthesis, and rain, fog, sun flare, sunny overexposure, and snow are generated respectively.

**Figure 3 animals-14-02800-f003:**
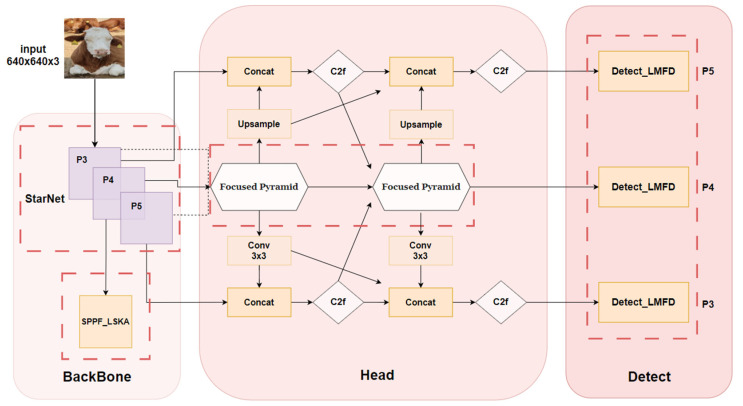
CBR-YOLO Network structure diagram.

**Figure 4 animals-14-02800-f004:**
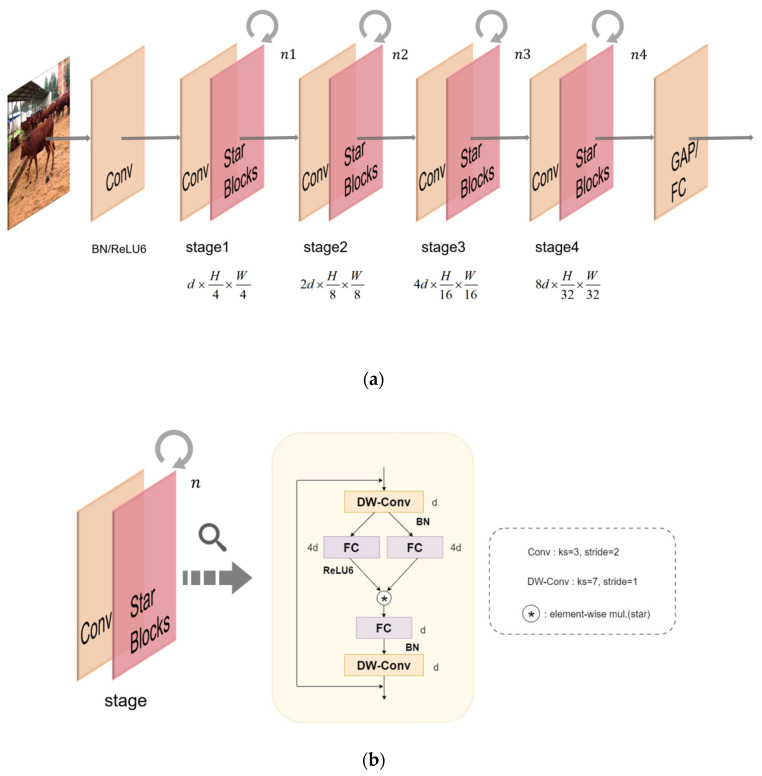
(**a**,**b**) represent an overview of StarNet architecture in this article.

**Figure 5 animals-14-02800-f005:**
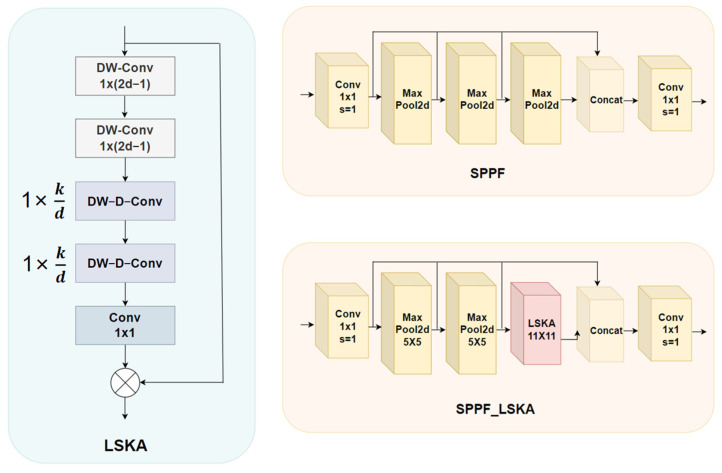
Large Separable Kernel Attention modules and SPPF are compared before and after modification.

**Figure 6 animals-14-02800-f006:**
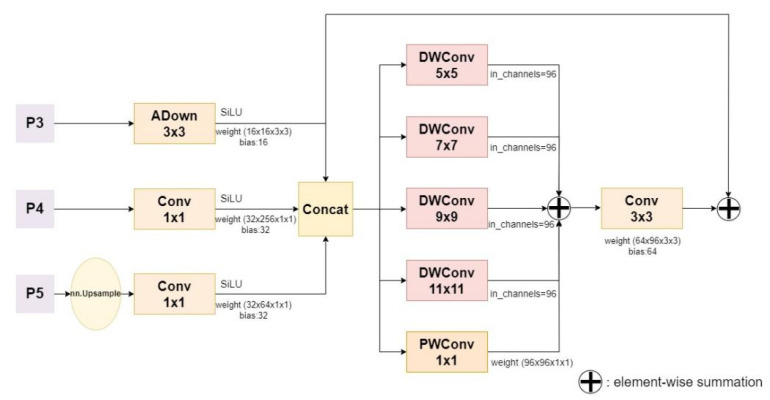
MCFP module architecture.

**Figure 7 animals-14-02800-f007:**
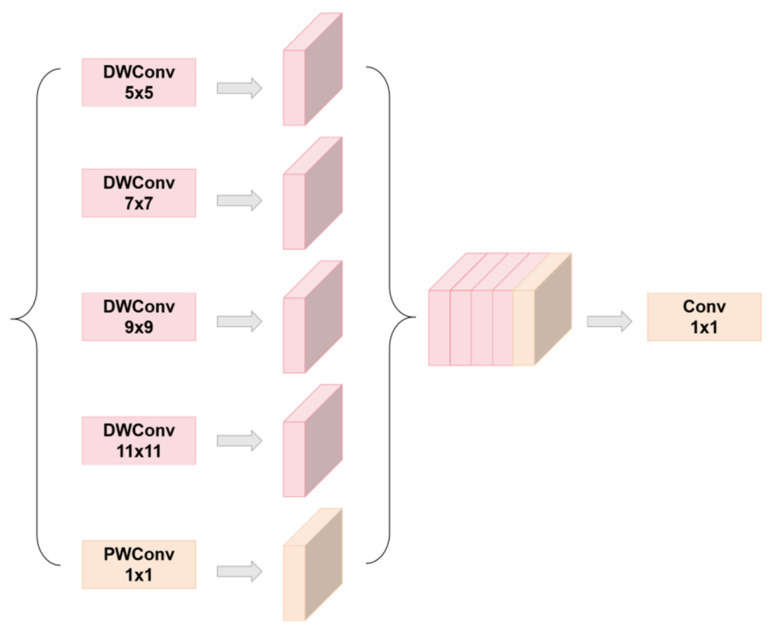
The process of receptive field feature fusion.

**Figure 8 animals-14-02800-f008:**
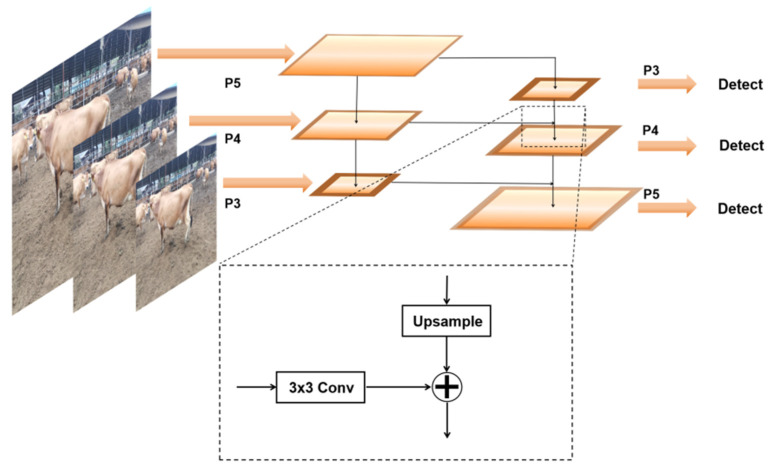
MCFP Modular multi-scale feature fusion process.

**Figure 9 animals-14-02800-f009:**
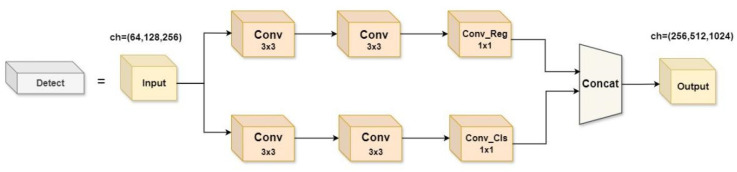
YOLOv8 Original detection head.

**Figure 10 animals-14-02800-f010:**
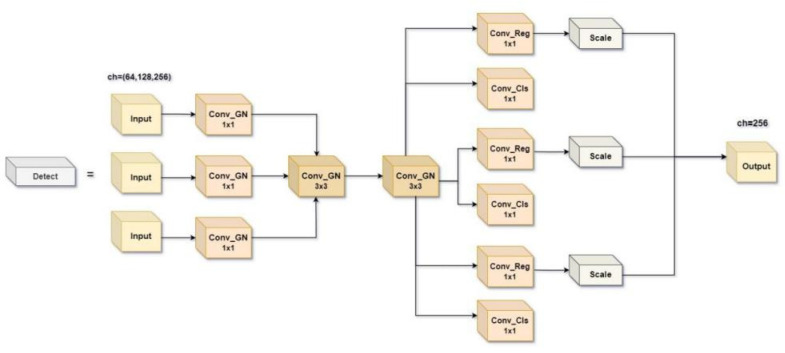
LMFD Detection head.

**Figure 11 animals-14-02800-f011:**
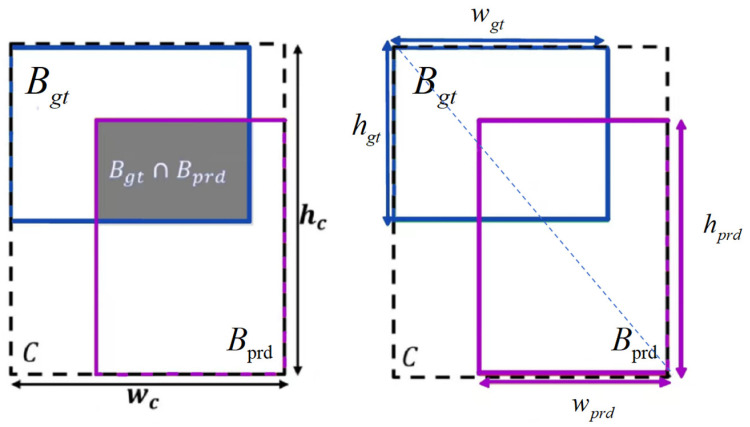
Existing bounding box regression metrics compute factors.

**Figure 12 animals-14-02800-f012:**
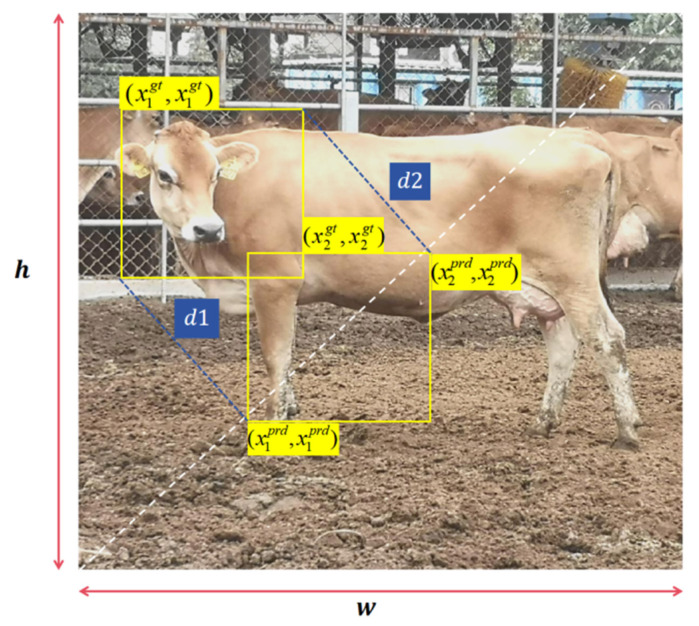
Factors influencing the IoU of Inner-MPD.

**Figure 13 animals-14-02800-f013:**
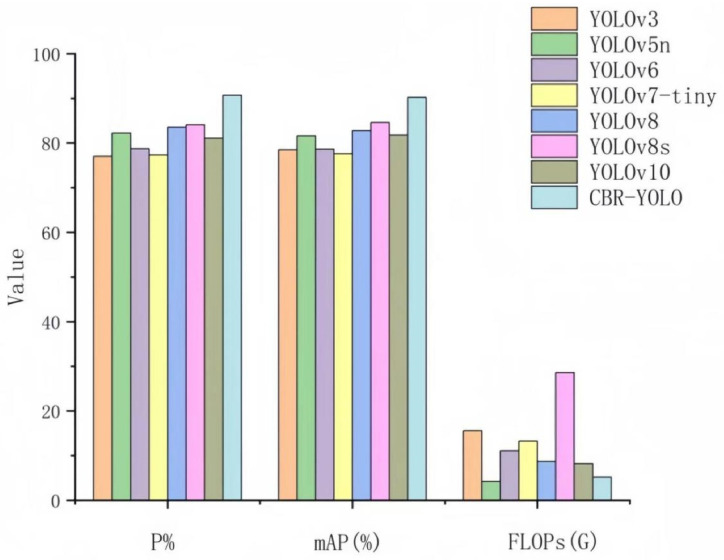
Comparison of metrics for eight YOLO algorithms.

**Figure 14 animals-14-02800-f014:**
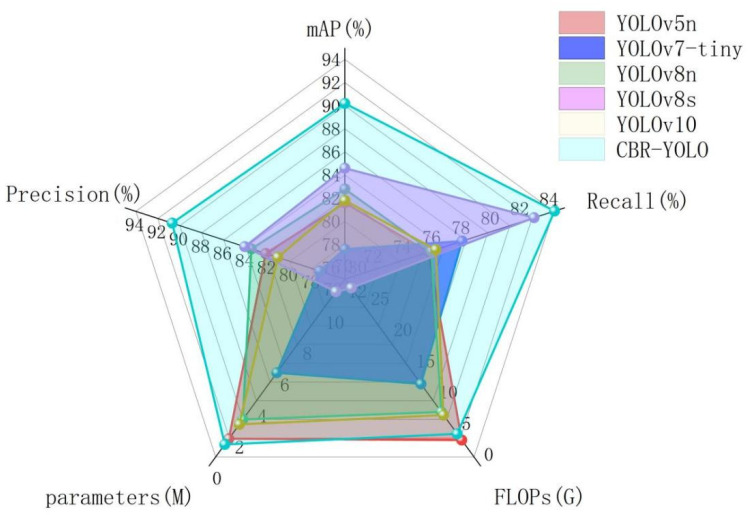
Comparison of Comprehensive Performance of Six Detection Algorithms.

**Figure 15 animals-14-02800-f015:**
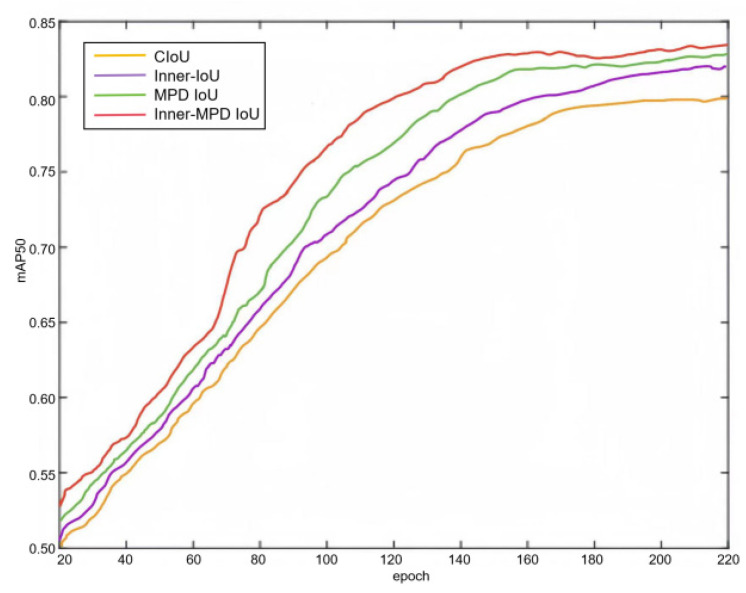
Comparison of four types of IoU loss performance.

**Figure 16 animals-14-02800-f016:**
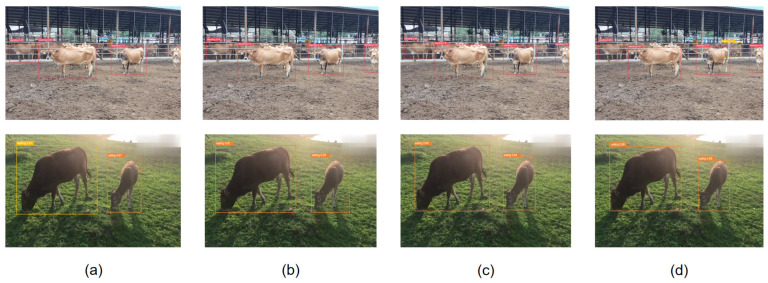
From column (**a**–**d**), they represent CIoU Loss in sequence, Inner-IoU Loss, MPDIoU Loss, The testing results of Inner MPD IoU Loss on this experimental dataset in this study.

**Figure 17 animals-14-02800-f017:**
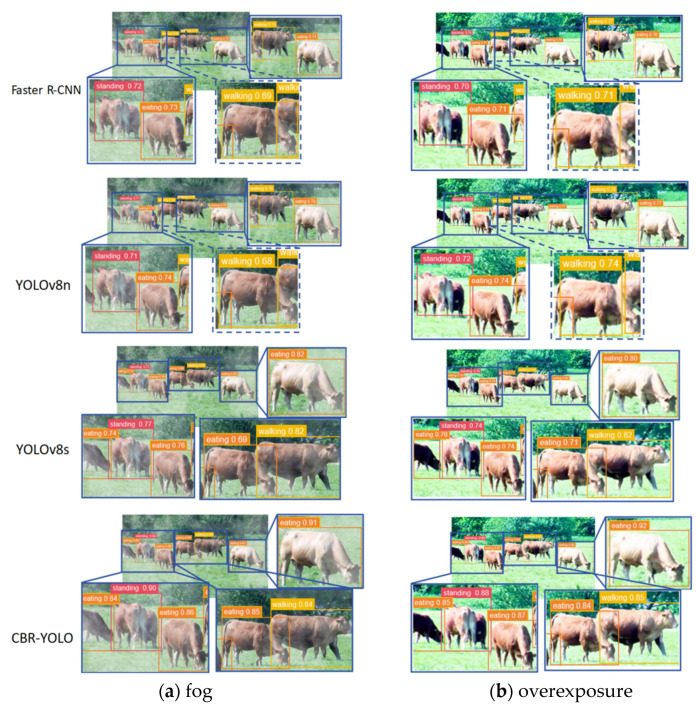
(**a**–**d**) represent the model recognition performance under four different weather conditions; from top to bottom they are Faster R-CNN, YOLOv8n, YOLOv8s, and CBR-YOLO.

**Figure 18 animals-14-02800-f018:**
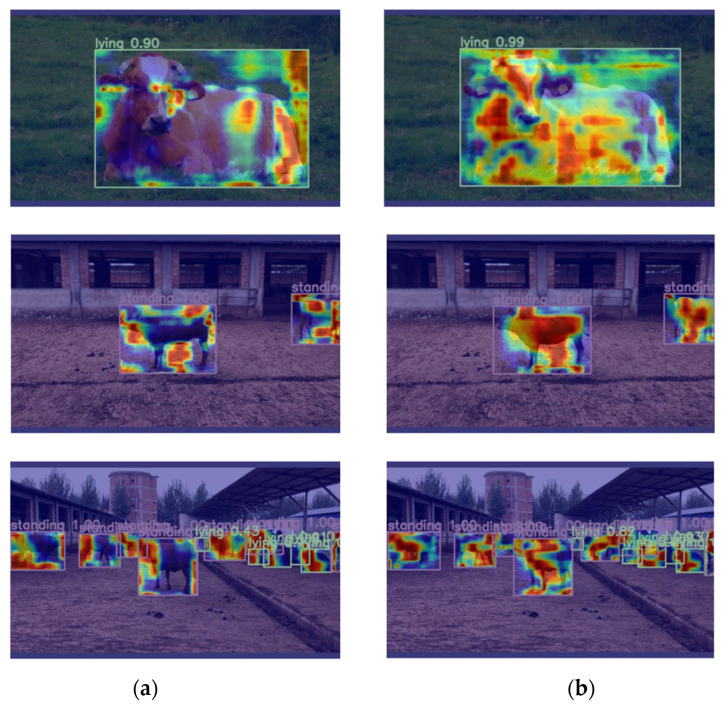
Comparison of heat maps before and after model optimization. Note: column (**a**) represents the YOLOv8n heatmap image, and column (**b**) represents the CBR-YOLO heatmap image.

**Figure 19 animals-14-02800-f019:**
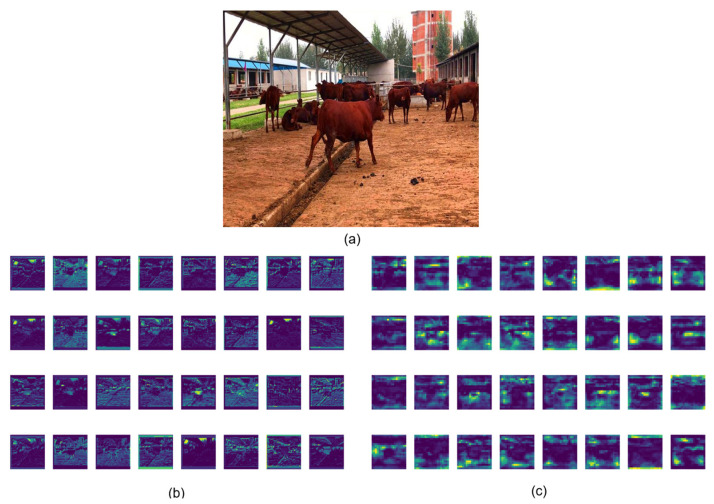
(**a**) is the original image, (**b**) is the feature map of StarNet in CBR-YOLO, (**c**) is the feature map of the first layer C2f of YOLOv8n.

**Table 1 animals-14-02800-t001:** Distribution of data.

	Image Quantity	Standing	Walking	Eating	Lying
Training set	3805	3159	3025	2561	2943
Validation set	621	554	315	279	312
Test set	625	571	343	230	337
All	5051	4284	3683	3070	3592

**Table 2 animals-14-02800-t002:** Experimental Environment Configuration.

Environment Configuration	Parameters
GPU	2*A100(80 GB)
CPU	Intel(R)Xeon(R)Gold 6148 CPU @2.40 GHz
Development environment	PyCharm 2023.2.5
Language	Python 3.8.10
Framework	PyTorch 2.0.1
Operating platform	CUDA 11.8
Operating system	Linux

**Table 3 animals-14-02800-t003:** Comparison of object detection results from different algorithms.

Models	P%	mAP%	Recall%	FLOPs/G	Parameters
SSD [18]	79.9	80.1	76.2	206.6	4.48 × 10^7^
Faster R-CNN [19]	82.8	82.0	82.7	310.7	2.47 × 10^7^
YOLOv3 [20]	77.0	78.5	76.1	15.6	8.67 × 10^6^
YOLOv3-tiny	76.9	65.1	68.8	12.9	6.93 × 10^6^
YOLOv5n [21]	82.2	81.6	75.7	4.2	1.76 × 10^6^
YOLOv6 [22]	78.7	78.6	69.0	11.1	4.23 × 10^6^
YOLOv7-tiny [23]	77.3	77.6	78.0	13.2	6.0 × 10^6^
YOLOv8n	83.5	82.8	75.9	8.7	3.00 × 10^6^
YOLOv8s	84.1	84.6	82.9	28.6	1.12 × 10^7^
YOLOv8m	85.7	85.1	83.9	78.9	2.59 × 10^7^
YOLOv9	81.4	80.9	76.8	26.7	6.0 × 10^7^
YOLOv10 [24]	81.1	81.8	76.2	8.2	2.69 × 10^6^
CBR-YOLO	90.7	90.2	84.3	4.8	1.40 × 10^6^

**Table 4 animals-14-02800-t004:** Accuracy of Four Representative Models under Different Behaviors of Cattle.

Models	Lying Precision (%)	Standing Precision (%)	Eating Precision (%)	Walking Precision (%)
Faster R-CNN	84.2	77.9	87.4	81.7
YOLOv8n	84.2	79.6	88.8	81.3
YOLOv8s	84.6	80.7	87.5	83.4
CBR-YOLO	91.2	86.5	95.5	89.5

**Table 5 animals-14-02800-t005:** mAP indicators of four representative models under different behaviors of cattle.

Models	Lying mAP (%)	Standing mAP (%)	Eating mAP (%)	Walking mAP (%)
Faster R-CNN	77.8	81.2	86.7	82.5
YOLOv8n	78.9	81.6	86.5	83.6
YOLOv8s	80.6	83.2	87.8	86.5
CBR-YOLO	86.9	88.9	93.4	91.5

**Table 6 animals-14-02800-t006:** Results of ablation experiments with different optimization modules.

Model	Inner-MPD IoU	StarNet	LSKA	MCFP	LMFD	mAP@0.5/%	Precision/%	Parameters/M	FLOPs/G
1						82.8	83.5	3.01	8.7
2	√					84.1	84.9	3.01	8.7
3	√	√				84.3	86.8	2.36	6.5
4	√	√	√			87.3	87.7	2.43	7.2
5	√			√		87.5	87.8	3.20	10.1
6	√		√	√		88.0	88.1	3.47	10.4
7	√	√		√		89.6	90.1	1.66	6.0
8	√	√	√	√		90.1	90.9	1.73	6.1
Ours	√	√	√	√	√	90.2	90.7	1.4	5.2

**Table 7 animals-14-02800-t007:** Comparison of LSKA modules at different positions.

StarNet	C2f	MCFP	SPPF	mAP@0.5 (%)	Precision/%
√				83.1	84.8
	√			82.9	84.1
		√		81.4	84.6
			√	84.7	85.9

**Table 8 animals-14-02800-t008:** Comparison of two detection head experiments.

	Params/M	FLOPs	Precision
Yolov8n_Detect	7.52 × 10^5^	8.9 G	83.50%
LMFD	1.12 × 10^5^	6.7 G	84.9%

## Data Availability

All new research data were presented in this contribution.

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
