# Peer review of "Research on the Behavior Recognition of Beef Cattle Based on the Improved Lightweight CBR-YOLO Model Based on YOLOv8 in Multi-Scene Weather"

_animals, 2024, doi:10.3390/ani14192800_

Round 1

Reviewer 1 Report (Previous Reviewer 2)

Comments and Suggestions for Authors

The manuscript was revised based on the review comments. 

Author Response

Dear reviewer, thank you so much for your reviewing! We deeply appreciate your recognition of our research work.

Reviewer 2 Report (New Reviewer)

Comments and Suggestions for Authors

Manuscript "Research on the behavior recognition of beef cattle based on the improved lightweight CBR-YOLO model based on YOLOv8 in multi-scene weather" is very interesting.

General comments:
Sections of the manuscript marked in yellow suggest that it is for the version after revisions suggested by earlier reviewers. Unfortunately, I did not receive the authors' response to these comments.

Authors presented a model based on improved YOLOv8, Cattle Behavior Recognition-YOLO (CBR-YOLO), which aims to accurately identify the behavior of cattle.
Authors simulated a variety of weather conditions and introduced multi-target detection technology to achieve comprehensive monitoring of cattle and their status.
At least that's what the authors assumed. Unfortunately, in the manuscript, the goals set were not realized.

Detailed comments:
The authors confuse the terms "simulation" with "generation." If a simulation study was actually conducted, the manuscript should be completed: indication of input data, values of assumed parameters changing in the analysis, providing a model on the basis of which dependent values of assumed independent variables are determined, providing formulas for estimation of simulated parameters, testing of these parameters.
The formulas on page 7 are written in a very sloppy manner.
Not all symbols are explained.
The figure captions do not explain the content of the figures.
Line 348: Multiplying by "100%" means multiplying by 1. Such notation makes no sense!

Mistakes that need to be corrected:
Line 41: ".."
L78: "(Jocher.2023)[9]"
L104, L159: "x", letter or symbol?
L108: "∗" This is not a multiplication sign!
Variables should be written in italics. Errors are throughout the manuscript.

Paper needs major revision.

Author Response

Dear reviewer, the following is our reply to your comments,

Comments 1: Sections of the manuscript marked in yellow suggest that it is for the version after revisions suggested by earlier reviewers. Unfortunately, I did not receive the authors' response to these comments.

Response 1: Thank you for pointing this out. I feel very sorry for this, which is my negligence. Now we have uploaded the reply to the reviewer to the system, sorry again.

Comments 2: Authors presented a model based on improved YOLOv8, Cattle Behavior Recognition-YOLO (CBR-YOLO), which aims to accurately identify the behavior of cattle.
Authors simulated a variety of weather conditions and introduced multi-target detection technology to achieve comprehensive monitoring of cattle and their status.At least that's what the authors assumed. Unfortunately, in the manuscript, the goals set were not realized.

Response 2: Thank you for pointing this out. In response to this problem, we are very sorry that we did not make it clear in the article. In terms of multi-objective technology, we mean that it can realize the recognition of multiple cattle behaviors in the picture, but it does not include identity recognition, which will be a new research in our subsequent experiments.I apologize for this misrepresentation, it has been modified in the main text. (p22 L515-523)

Comments 3: The authors confuse the terms "simulation" with "generation." If a simulation study was actually conducted, the manuscript should be completed: indication of input data, values of assumed parameters changing in the analysis, providing a model on the basis of which dependent values of assumed independent variables are determined, providing formulas for estimation of simulated parameters, testing of these parameters.

Response 3: Thank you very much for pointing out this error. We agree with this comment. We have modified the simulation in the article to generate.For the weather synthesis technology, we adopted A linear mixing method, and weighted the original image by randomly generating brightness parameters A and transmittance t to achieve the weather generation effect. These operations are implemented through python code.The intensity of the generated weather we choose in Figure 7 is random and not limited to weak or strong, and the advantages of our model can be fully demonstrated by experiments.Thanks again.

(p1 L31,p4 L137,p4 L126-129)

Comments 4: The formulas on page 7 are written in a very sloppy manner. Not all symbols are explained.

Response 4: Thank you very much for pointing out this error. We agree with this comment. Now we have made additions in the body to ensure that each variable is explained in the article.(p7 L194-195)

Comments 5: The figure captions do not explain the content of the figures.

Response 5: Thank you for pointing this out. We agree with this comment.Now I have corrected this error.(p6 L189-191)

Comments 6: Line 348: Multiplying by "100%" means multiplying by 1. Such notation makes no sense.

Response 6: Thank you for pointing this out.But I want to explain multiplying by 100% in the Recall calculation formula is to express the result as a percentage, making it easier to understand and compare.  This operation does not change the essential meaning of the formula, but simply provides a more intuitive presentation of the results.

Comments 7: Line 41: ".."

Response 7: Thank you for pointing this out. We agree with this comment.Now I have corrected this error.(p1 L41)

Comments 8: L78: "(Jocher.2023)[9]"

Response 8: Thank you for pointing this out. We agree with this comment.Now I have corrected this error.(p2 L77)

Comments 9: L104, L159: "x", letter or symbol?

Response 9: Thank you for pointing this out. Now I have corrected this error.

Comments 10: Variables should be written in italics.

Response 10: Thank you for pointing this out. Now I have corrected this error.

With best wishes

Reviewer 3 Report (New Reviewer)

Comments and Suggestions for Authors

The manuscript presents a novel approach to cattle behaviour recognition using an improved YOLOv8 model, dubbed CBR-YOLO. The authors introduce several innovations, such as the Inner-MPD IoU Loss function, the Multi-Convolutional Focused Pyramid (MCFP) module, and the Lightweight Multi-scale Feature Fusion Detection Head (LMFD). These innovations are aimed at enhancing the model's ability to detect and classify cattle behaviours under various weather conditions while maintaining computational efficiency, making it suitable for deployment on resource-limited edge devices. The article needs minor revision. There are some points that the authors need to consider:

1-The authors compare their model to several others; the discussion could be improved by providing more insights into why certain models performed better or worse in specific scenarios. 

2- The research mainly focuses on cattle behaviour recognition under various weather conditions. However, it would be beneficial to discuss the potential for generalizing this approach to other types of livestock or different environmental conditions.

3-Include a discussion of the current approach's limitations and possible avenues for future work, such as the challenges of scaling this model to larger herds or different species.

4-Improve the discussion section to include more analysis of the comparative results and the potential generalization of the model to other applications.

Author Response

Dear reviewer, thank you so much for your comments! We appreciate your recognition of our research work.

Comments 1: The authors compare their model to several others; the discussion could be improved by providing more insights into why certain models performed better or worse in specific scenarios. 

Response 1: Thank you for pointing this out. We agree with this comment. We have revised the original text. Thank you for pointing out the problem. (p22 L510-514)

Comments 2: The research mainly focuses on cattle behaviour recognition under various weather conditions. However, it would be beneficial to discuss the potential for generalizing this approach to other types of livestock or different environmental conditions.

Response 2: Thank you for pointing this out. We agree with this comment. In the future, we will use transfer learning to identify other animal behaviors, and we also hope to further study the identification of animal identities, which I have added in the discussion.(p22 L515-530)

Comments 3:A discussion of the limitations of current methods and possible avenues for future work was included, such as the challenges of extending this model to larger herds or different species.

Response 3:Thank you for pointing that out. We agree with this comment. We revised the end of the article accordingly.

Comments 4:The discussion section has been improved to include more analysis of the comparison results and potential generalization of the model to other applications.

Response 3:Thank you for pointing that out. We agree with this comment. We revised the end of the article accordingly. Page 22 L515-523)

Best wishes

Round 2

Reviewer 2 Report (New Reviewer)

Comments and Suggestions for Authors

The manuscript still needs minor revisions:

Line 129 (and throughout the manuscript): * is not a multiplication symbol; this should be corrected,
Line 130: 'x' symbol not letter,
Comments 4: The formulas on page 7 are written in a very sloppy manner. Not all symbols are explaine. Still not corrected, although the authors' response indicates this!
Comments 5: The figure captions do not explain the content of the figures. Still not corrected, although the authors' response indicates this!
Comments 6: Line 348: Multiplying by "100%" means multiplying by 1. Such notation makes no sense. If we want to express a value as a percentage, we multiply it by 100 and write down the percentage symbol. Multiplying by "100%" is the same as multiplying by "1". Please, for verification, make a simple test in Excel: =3*100%. The result will be equal to 3.

This manuscript is a resubmission of an earlier submission. The following is a list of the peer review reports and author responses from that submission.

Round 1

Reviewer 1 Report

Comments and Suggestions for Authors

General comments

This manuscript describes a novel deep network architecture for the identification of cattle postures from images. I have several questions that I would like the authors to address.

The article is presented as describing an improvement of the YOLO method for the detection of beef cattle behaviour. However, the precise aims of the work are not clearly written. Many aspects are covered, such as changes to the loss function, the evaluation of weather conditions on the performance of detection… but their place in the overall picture is not entirely clear to me. The authors should clearly state what they intend to evaluate: accuracy of detection, speed… and focus the paper on these research questions.

In the introduction and in various places in the manuscript, the lack of familiarity of the authors with cattle farming and the possible applications of their work is apparent:

-        The YOLO algorithm is based on static images. It can only detect postures, not behaviours. The paper is on the evaluation of a method for the detection of cattle postures.

-        If the proposed method works, what should be possible is to detect cattle postures in real time from videos. Using this for practical purposes would require further steps such as creating time budgets.

-        The authors should provide more accurate perspectives on the above points from the literature.

-        What does the word biomanufacturing (p1 L51) mean?

-        The cows in Figure 1 are dairy cows (Jersey breed), not beef cows.

-        The animal in Figure 19 is not cattle. It is a buffalo.

The construction of the train, validation and test datasets is favourable to the method, because cows in the same environment will likely be present in all datasets. It would have been preferable to evaluate the methods with making sure that cows in the same farm or environment were note present in all the datasets. A better approach would be to have an entire dataset (one of the three you selected for your study) as a test set.

Some figures are only slightly modified from previous publications. This should be written in figure captions. For example: Figure 4 is adapted from reference 10. Figures 11 and 12 are adapted from reference 15.

The authors claim that they want to evaluate the impact of various weather conditions on the performance of their method. The different weather conditions are created by applying an algorithm on the pictures. The relevance and performance of this method to achieve the stated goal are impossible to determine from what is presented.

The new loss function would warrant a full paper, not be included as part of this one. Theoretical justifications for the new method should be provided and the function should be rigorously evaluated as was done by Ma and Xu. It seems like the function presented in the equation number 8 does not represent a loss function that would be minimised during training. Furthermore, equations 2 to 5 are supposed to represent the output of SPFF_LSKA, but are straight copy pasted from the LSKA paper (with typos), and represent the output of the LSKA block. The output should be different as it also involves the concatenation with the two pooling layers before.

How were the models against which the new model is evaluated selected? For example, why weren’t the YOLOv8m, YOLOv8l as well as the YOLOv8x, which would probably have performed better than YOLOv8-N not included?

Figures 16 and 17 are hard to read. Why is there a reference to cows in oestrus? This was not part of the behaviours of interest.

In all table captions, replace ‘Tabel’ with ‘Table’. Similarly, the figures should be either entitled with Fig. number or Figure number.

In Table 2, rewrite the column headers in English.

Author Response

Comments 1: The article is presented as describing an improvement of the YOLO method for the detection of beef cattle behaviour. However, the precise aims of the work are not clearly written. Many aspects are covered, such as changes to the loss function, the evaluation of weather conditions on the performance of detection… but their place in the overall picture is not entirely clear to me.

Response 1: Thank you for pointing this out. We agree with this comment. Therefore, we have made the following changes: This study aims to provide a lightweight and accurate behavior detection model that can adapt to real weather changes and meet the requirements for future deployment on edge devices. Each module design in this paper is intended to meet these needs by reducing parameters and minimizing redundancy.(p1 L28-32,p2 L81-95)

Comments 2: The YOLO algorithm is based on static images. It can only detect postures, not behaviours. The paper is on the evaluation of a method for the detection of cattle postures.

Response 2: Thank you for pointing this out. YOLOv8 algorithm has been able to detect videos, and this paper also selected ten videos as verification, which can meet the needs of behavior recognition. The text has been cited to assist. (p2 L78-80)

Comments 3: What does the word biomanufacturing (p1 L51) mean?

Response 3Thank you for pointing this out. I apologize for this misrepresentation, my intention was to promote sustainable development in areas such as agricultural science and technology, and it has now been amended in the text. (p2 L48)

Comments 4: The cows in Figure 1 are dairy cows (Jersey breed), not beef cows.

Response 4: Thank you for taking the time to review and point this out. But we have confirmed that the data set was collected at a large beef cattle farm in Changchun, Jilin Province, China, and have just contacted the staff of the farm to confirm that the cattle are indeed Holstein beef cattle, not Jersey cows.

Comments 5: The animal in Figure 19 is not cattle. It is a buffalo.

Response 5Thank you very much for pointing out this error, we apologize for this error and have removed this graph from the dataset and replaced the experimental image.(p21 Fig,19)

Comments 6: The construction of the train, validation and test datasets is favourable to the method, because cows in the same environment will likely be present in all datasets. It would have been preferable to evaluate the methods with making sure that cows in the same farm or environment were note present in all the datasets. A better approach would be to have an entire dataset (one of the three you selected for your study) as a test set.

Response 6 : Thanks to the reviewer for pointing out this problem. In building the dataset, we have ensured that cows in the same environment are present in all datasets, but this is not highlighted in the paper. (p3 L118)

Comments 7: The authors claim that they want to evaluate the impact of various weather conditions on the performance of their method. The different weather conditions are created by applying an algorithm on the pictures. The relevance and performance of this method to achieve the stated goal are impossible to determine from what is presented.

Response 7: Thank you for pointing this out. Due to the unpredictability of the weather in the real environment, previous studies were limited to sunny weather, which greatly limited the applicability of the model in real life. Real weather is difficult to capture accurately, so we use Python algorithms to synthesize weather conditions and simulate real weather. This method not only approximates the actual agricultural environment, but also enhances the robustness of the model. The new changes are reflected in the paper. (p3 L123-124, p4 Figure 2)

Comments 8: Equations 2 to 5 are supposed to represent the output of SPFF_LSKA, but are straight copy pasted from the LSKA paper (with typos), and represent the output of the LSKA block. The output should be different as it also involves the concatenation with the two pooling layers before.

Response 8: Thank you for pointing this out. We agree with you and have revised and explained this issue.(p7 L193-204)

Comments 9: How were the models against which the new model is evaluated selected? For example, why weren’t the YOLOv8m, YOLOv8l as well as the YOLOv8x, which would probably have performed better than YOLOv8-N not included?

回应 9谢谢你指出这一点。 我们必须承认,在以前的研究中,yolov8m 和 yolov8l 的准确率会高于 yolov8n,因为它们具有更大的模型容量,可以学习更复杂的对象特征和上下文信息。但是,它们具有太多的计算和参数数量,无法满足本研究中轻量级的需求。我们添加了 experimented with yolov8m (第 13 页 L354-356,表 3)

评论 10: 图 16 和图 17 很难阅读。为什么在发情时提到了奶牛?这不是感兴趣的行为的一部分。

响应 10感谢审阅者指出此问题。 我为这种虚假陈述道歉,它已在正文中进行了修改。第 16 页 L394-395)

Comments 11: In all table captions, replace ‘Tabel’ with ‘Table’. Similarly, the figures should be either entitled with Fig. number or Figure number.

Response 11: Thanks to the reviewer for pointing out this problem. I apologize for this misrepresentation, now all the tables have been corrected.

Comments 12: In Table 2, rewrite the column headers in English.

Response 12: Thanks to the reviewer for pointing out this problem. Now I have corrected this error.(p12 Table 2)

Reviewer 2 Report

Comments and Suggestions for Authors Summary: The manuscript proposed an improved lightweight object detection model aims to provide accurate behavior recognition (standing, walking, eating, and lying down) with complex weather conditions, occlusions, and varying scales of cattle postures, making it suitable for deployment on resource-limited devices in farm environments. The authors introduce several novel components to the model, including:(1)Inner MPDIoU Loss: A new loss function that combines Inner-IoU Loss and MPDIoU Loss to improve bounding box regression and handle occlusions and dense distributions.(2)Multi-Convolutional Focused Pyramid (MCFP) module: A novel module that integrates multi-scale context information and fine-grained details of cattle in different states.(3)Lightweight Multi-scale Feature Fusion Detection Head (LMFD): A lightweight detection head that fuses multi-scale features and reduces computational cost and parameters. Comparing with other 12 models, the method are better at addressing real-world challenges like varying weather and lighting changes through data augmentation and architectural innovations. Furthermore, the method achieves 90.2% average accuracy, while reducing 3.9G floating-point operations.    Comments: (1) The data quality seems adequate and the analysis have been clear presented in tables and figures. However, more details on the data collection process, annotation procedures, and datasets distribution (e.g., class imbalance, occlusion levels, etc.) could be helpful for understanding potential biases. (2) The manuscript could benefit from a more detailed discussion of potential limitations and future research directions, such as exploring other types of cattle behaviors, handling dynamic scenes, or extending the approach to other livestock species. (3) The authors could consider providing more implementation details and methodology of hyperparameter settings to facilitate reproducibility and enable other researchers to build upon their work. (4) The manuscript could be improved by providing more qualitative examples and visualizations of the model's performance on challenging cases, such as occluded or densely distributed cattle, to better illustrate the effectiveness of the proposed methods. (5) Although the references provided in the paper are relevant and appropriate for the subject area, the authors could consider including more recent references in the field of object detection and behavior recognition, particularly those related to lightweight models or methods for handling complex environments.   Based on the technical novelties, promising results, and potential practical impact, the manuscript presents an interesting and relevant contribution to the field of cattle behavior recognition, addressing a practical problem with a novel and efficient approach. While there are some areas for potential improvement and further exploration, I would recommend accepting this manuscript for publication after addressing the comments and suggestions provided above.

Author Response

Comments 1: Inner MPDIoU Loss: A new loss function that combines Inner-IoU Loss and MPDIoU Loss to improve bounding box regression and handle occlusions and dense distributions.

Response 1: Thank you for pointing this out. We agree with this comment. We have revised the original text. Thank you again for your patient review (p2 2.1.2 Data set construction)

评论 2: 对手稿可能受益于对潜在局限性和未来研究方向的更详细讨论,例如探索其他类型的牛行为、处理动态场景或将方法扩展到其他牲畜物种。

响应 2:谢谢你指出这一点。我们同意这个评论。我们修改了原文的讨论部分。(第 22 页 L510-513)

Reviewer 3 Report

Comments and Suggestions for Authors

1. The “Simple Summary” needs to be rewritten.

2. The entire manuscript lacks tense agreement. Past tense should be used for any research and experiments that have been performed in the past.

3. There are a lot of grammar errors and other writing style issues throughout the manuscript that should be corrected. Please perform extensive English language editing.  

4. There are a lot of unexplained abbreviations.

5. The weather synthesis is a questionable method. Adding a little blur to simulate fogs (Fig. 2c) or several white flakes to simulate the snow (Fig. 2c again? Should be Fig. 2f.) makes little if not no changes to the features of pictures.

6. The overall model improvement method in Fig. 3 needs explanation in detail.

7. The parameters in Fig. 4a need to be explained.

8. Why is Fig. 4 divided into two sub-figures while they are not mentioned separately in the text body?

9. The asterisk is used for element-wise multiplication sometimes, while as convolution or numerical product elsewhere. Please be consistent.

10. Equations 2-3, 6, and 7 are not described or explained.

11. Figures 6, 10, and 19 are difficult to read.

12. Foreign language appears in “Tabel” 2.

13. Although the manuscript contains a lot of figures and equations, the originality is limited. The quality and the authenticity of the datasets are low, especially considering the questionable method to manipulate pictures instead of capturing photos of real weather conditions. The results and the conclusion are thus questionable.

Comments on the Quality of English Language

Please see the comments above.

Author Response

For review article

Response to Reviewer 3 Comments

1. Summary

2. Questions for General Evaluation

Reviewer’s Evaluation

Response and Revisions

Does the introduction provide sufficient background and include all relevant references?

Yes/Can be improved/Must be improved/Not applicable

Are all the cited references relevant to the research?

Yes/Can be improved/Must be improved/Not applicable

Is the research design appropriate?

Yes/Can be improved/Must be improved/Not applicable

Are the methods adequately described?

Yes/Can be improved/Must be improved/Not applicable

Are the results clearly presented?

Yes/Can be improved/Must be improved/Not applicable

Are the conclusions supported by the results?

Yes/Can be improved/Must be improved/Not applicable

Comments 1: The “Simple Summary” needs to be rewritten.

Response 1: Thank you for pointing this out. We agree with this comment. We have revised the original text. Thank you for pointing out the problem. (p1 L20-27)

Comments 2: The entire manuscript lacks tense agreement. Past tense should be used for any research and experiments that have been performed in the past.

Response 2: Thank you for pointing this out. We agree with this comment. We have adjusted the tense grammar of this paper. Thanks again for your careful reading.

Comments 3: There are a lot of grammar errors and other writing style issues throughout the manuscript that should be corrected. Please perform extensive English language editing.

Response 3: Thank you for pointing this out. We agree with this comment. There were indeed many grammatical problems in the last edition of the manuscript, and we have now revised and polished this article.

Comments 4: There are a lot of unexplained abbreviations.

Response 3: Thank you for pointing this out. We agree with this comment. We have annotated all abbreviations as they first appear in the new manuscript. Thank you for your criticism

Comments 5: The weather synthesis is a questionable method. Adding a little blur to simulate fogs (Fig. 2c) or several white flakes to simulate the snow (Fig. 2c again? Should be Fig. 2f.) makes little if not no changes to the features of pictures

Response 5: Thank you for pointing this out. Due to the unpredictability of the weather in the real environment, previous studies were limited to sunny weather, which greatly limited the applicability of the model in real life. Real weather is difficult to capture accurately, so we use Python algorithms to synthesize weather conditions and simulate real weather. This method not only approximates the actual agricultural environment, but also enhances the robustness of the model. In the paper, we wrote that all the changes in the weather intensity and the location of the weather are random, and this step occurs in the process of preprocessing the data set before the program pre-training. Figure 2 is an example of a composite weather event of moderate intensity. The new changes are reflected in the paper. (p3 L123-124, p4 Figure 2)

Comments 5: The overall model improvement method in Fig. 3 needs explanation in detail.

Response 5: Thank you for pointing this out. We agree with this comment. I have modified Figure 3 and marked all the improvements with red dotted boxes. Section 2.2.1 describes the overall improvement strategy for CBR-YOLO, and each subsequent section details each improvement. (p2 L147-148,p3 Fig.3)

Comments 6: The parameters in Fig. 4a need to be explained.

Response 6: Thank you for pointing this out. We agree with this comment. I did neglect this point before, and now I have revised it. Thank you again for your patient review. (p5 L156-163)

Comments 7: Why is Fig. 4 divided into two sub-figures while they are not mentioned separately in the text body?

Response 7: Thank you for pointing this out. We agree with this comment. We reorganized the language and changed it accordingly. (p5 L156-163)

Comments 8: Equations 2-3, 6, and 7 are not described or explained.

Response 8: Thank you for pointing this out. We agree with this comment. In the previous writing, we did neglect this point, and now we have added to the main body. (p7 L198-204,p11 L302-307)

Comments 8: Foreign language appears in “Tabel” 2.

Response 8: Thank you for pointing this out. We agree with this comment. I apologize for this misrepresentation, now all the tables have been corrected. Thank you again for pointing out the problem.

4. Response to Comments on the Quality of English Language

响应:对于上一篇文章中的语法错误,我们深表歉意。现在我们已经完善了论文,并感谢审稿人的细心建议。所有校正均以黄色高亮显示。